# Treatment of Donor Cells with Oxidative Phosphorylation Inhibitor CPI Enhances Porcine Cloned Embryo Development

**DOI:** 10.3390/ani14091362

**Published:** 2024-04-30

**Authors:** Jinping Cao, Yazheng Dong, Zheng Li, Shunbo Wang, Zhenfang Wu, Enqin Zheng, Zicong Li

**Affiliations:** 1National Engineering Research Center for Breeding Swine Industry, South China Agricultural University, Guangzhou 510642, China; caojinping2021@scau.edu.cn (J.C.); dongyazheng@stu.scau.edu.cn (Y.D.); lizheng2022@stu.scau.edu.cn (Z.L.); 03028301@stu.scau.edu.cn (S.W.); wzfemail@163.com (Z.W.); 2State Key Laboratory of Swine and Poultry Breeding Industry, South China Agricultural University, Guangzhou 510642, China; 3National and Local Joint Engineering Research Center for Livestock and Poultry Breeding Industry, South China Agricultural University, Guangzhou 510642, China; 4Department of Animal Genetics, Breeding and Reproduction, College of Animal Science, South China Agricultural University, Guangzhou 510642, China; 5Guangdong Provincial Key Laboratory of Agro-Animal Genomics and Molecular Breeding, South China Agricultural University, Guangzhou 510642, China; 6Gene Bank of Guangdong Local Livestock and Poultry, South China Agricultural University, Guangzhou 510642, China

**Keywords:** SCNT, CPI, oxidative phosphorylation, glycolysis

## Abstract

**Simple Summary:**

The low developmental efficiency of cloned pig embryos limits the application of the pig cloning technique. In this study, a small molecule drug called CPI was added to the culture medium of nuclear donor cells to enhance the developmental capacity of subsequent cloned pig embryos. CPI treatment of nuclear donor cells changed the cellular energy metabolism status from oxidative phosphorylation to glycolysis, and significantly improved the developmental competence of subsequent cloned pig embryos. This study provides a simple approach to facilitate the development and application of pig cloning technology.

**Abstract:**

Somatic cell nuclear transfer (SCNT) technology holds great promise for livestock industry, life science and human biomedicine. However, the development and application of this technology is limited by the low developmental potential of SCNT embryos. The developmental competence of cloned embryos is influenced by the energy metabolic status of donor cells. The purpose of this study was to investigate the effects of CPI, an oxidative phosphorylation inhibitor, on the energy metabolism pathways of pig fibroblasts and the development of subsequent SCNT embryos. The results showed that treatment of porcine fibroblasts with CPI changed the cellular energy metabolic pathways from oxidative phosphorylation to glycolysis and enhanced the developmental ability of subsequent SCNT embryos. The present study establishes a simple, new way to improve pig cloning efficiency, helping to promote the development and application of pig SCNT technology.

## 1. Introduction

Somatic cell nuclear transfer (SCNT), also called cloning, enables the development of reconstructed embryos into complete individuals possessing identical nuclear genetic material to that of their donor cells. This technology holds significant application value in animal husbandry, life sciences, and human biomedicine. SCNT can be used to propagate excellent breeding stock, including top bulls [1], Duroc boars [2], Pietran boars [3], and help to save endangered animals such as European argali [4], African wildcats [5], and eastern Guangdong black pigs. SCNT is also employed to generate various genetically modified animals for use as human disease models [6,7], human xenotransplantation organ donors [8,9], and drug synthesis bioreactors [10,11]. Moreover, SCNT technology facilitates the generation of nuclear transfer embryonic stem cells (ntESC) for applications in human therapeutic cloning [12,13]. 

Nevertheless, the current efficiency of SCNT technology remains very low. Specifically, the development of pig SCNT embryos into blastocysts under in vitro culture conditions achieves an efficiency of approximately 20–30% [14,15,16], while in cattle, this efficiency ranges from 20% to 50% [17,18,19]. Furthermore, the full-term developmental rate of cloned pig embryos after transplantation into the reproductive tract of surrogate mothers is approximately 0.5–3% (calculated as the number of cloned animals born divided by the number of 1–2 cell stage cloned embryos transplanted) [20]. Cattle SCNT embryos exhibit a slightly higher efficiency, with a birth rate of about 10% (calculated as the number of cloned animals born divided by the number of blastocyst stage embryos transplanted) [21]. These considerably low developmental rates pose limitations to the development and application of SCNT technology. 

The energy utilization status of donor cells affects the subsequent development of SCNT embryos. Different cell types rely on different metabolic pathways to generate energy. Embryonic stem cells (ESCs) induced pluripotent stem cells (iPSCs), pluripotent stem cells (PSCs), and highly proliferative cells (such as endothelial cells, epithelial cells, and immune cells) predominantly depend on aerobic glycolysis for energy supply [22,23]. Notably, cells in early embryos, being stem cells, also primarily employ glycolysis to generate ATP for cell survival and proliferation [24]. In contrast, other types of somatic cells typically utilize mitochondrial oxidative phosphorylation as the primary source of energy production [25]. Evidence indicates that the transition of cellular energy metabolism pathways from oxidative phosphorylation to glycolysis is essential for reverting cells to an undifferentiated state and maintaining stemness [26]. Comparative analyses of the transcriptome of human iPSCs and fibroblasts reveal upregulation of glycolysis-related enzymes, and a shift of energy metabolism towards glycolysis in iPSCs [27]. The process of reprogramming mouse fibroblasts into totipotent stem cells also involves a switch from oxidative phosphorylation to glycolysis, resulting in reduced cellular oxygen consumption and increased lactate production [28]. 

The alterations in energy metabolism pathways are typically concomitant with the reprogramming of cell fates [29]. The changes in donor cell metabolism pathways have the potential to enhance the subsequent developmental efficiency of cloned pig embryos. A study has demonstrated that buffalo fetal fibroblasts with elevated expression levels of glycolytic enzyme genes exhibit a higher cloning efficiency [30]. Furthermore, treating buffalo fetal fibroblasts with glycolysis inducer PS48 resulted in the increased production of intracellular lactic acid and a consequential enhancement in the development of cloned embryos [30]. The treatment of porcine fibroblasts with 100 µM CoCl_2_ (a glycolysis inducer) led to the increased expression of glycolytic enzyme genes and the enhanced development of cloned embryos [31].

To investigate the influence of modifying donor cell energy metabolism pathways on the subsequent development of pig SCNT embryos, this study employed the oxidative phosphorylation inhibitor CPI to treat porcine fetal fibroblasts (PFFs), donor cells for cloned embryos. The energy metabolic status and global gene expression pattern were compared between CPI-treated and solvent-incubated negative control (NC) PFFs. The developmental competence of SCNT embryos generated from CPI-treated and NC PFFs was also compared.

## 2. Materials and Methods

### 2.1. Porcine Fetal Fibroblast Culture and Passaging

PFFs were obtained from Livestock Germplasm Resource Center, South China Agricultural University (Catalog No. B13XEHTF020200109, small-eared flower breed). The cells were cultured in a 37 °C, 5% CO_2_ thermostatic incubator. Upon achieving a cell growth confluence of 80~90%, the cells were ready for passaging. A 1 mL solution containing 0.25% Trypsin-EDTA was added, and the cells were incubated for 2 min. The digestion process was terminated by adding 2 mL of complete medium. Subsequently, centrifugation at 800 rpm for 5 min was performed, the supernatant was discarded, and the cells were transferred to two 100 mm culture dishes for ongoing cultivation.

### 2.2. CPI Treatment of Donor Cells

The concentration of CPI used for incubating donor cells was chosen according to a previously reported study [32]. Five mg of CPI (MedChem Express Biotechnology, Inc., Monmouth Junction, NJ, USA, item number 95809-78-2) was dissolved in 1.2867 mL of DMSO, preparing a mother liquor with a final concentration of 10 mM. Subsequently, aliquots of 10 μL, 50 μL, and 100 μL of the CPI mother liquor were added to 10 mL of the DMEM complete medium; the final concentrations of the resulting working solution were 10 μM, 50 μM and 100 μM, respectively. 

### 2.3. Reverse Transcription and qPCR

PFF RNA was extracted utilizing the conventional Trizol method, and the reverse transcription was prepared using Evo M-mLV Reverse Transcription Premix Kit II (Guangzhou Ruizhen Biotechnology Co., Ltd., Guangzhou, China). ChamQ Universal SYBR qPCR Master Mix kit (Nanjing Novelty Biotechnology Co., Nanjing, China) was used for qPCR. Primers and their sequences are shown in Table 1. PCR reactions were performed using a Quant Studio 7 Flex system (Thermo Fisher Scientific, Waltham, MA, USA) according to the parameters in Table 2. The qPCR was performed following a previously reported study [33]. The relative expression of genes was calculated using the 2^−∆∆Ct^ method. The data of each group were from three independent samples and each sample was measured with three technical replicates. A melting curve analysis was added at the end of the amplification procedure to confirm the specificity of the amplification.

### 2.4. TMRE Assay of Mitochondrial Membrane Potential (MMP)

The culture medium of PFFs treated with CPI for 48 h was aspirated, cells were washed with PBS, and a working solution was prepared in accordance with TMRE: serum-free medium = 1:1000 using the TMRE kit (Shanghai Beyotime Biotech, Shanghai, China, Item No. C2001S). Subsequently, 1 mL of the working solution was added to each well and incubated for 45 min in a 37 °C incubator, shielded from light. At the end of the incubation period, the supernatant was aspirated, and the cells were washed twice with a pre-warmed cell culture solution. Following this, 500 μL of the pre-warmed cell culture solution containing serum and phenol red was added. The observation and documentation of cell fluorescence were conducted under a fluorescence microscope, and images were quantified using ImageJ (13.0.6).

### 2.5. Measurement of Lactic Acid Content

A lactic acid content kit purchased from Shanghai Biyuntian Biotechnology Co., Ltd., Shanghai, China, (Item No.: C0017) was used for measuring lactic acid content. The extraction process involved a cell-to-extract ratio of 500~1000:1. Specifically, 1 mL of extract was added to the designated volume of cells, and the cells were subjected to ultrasonic ice bath treatment for cell lysis at a power of 300 W, with ultrasonication intervals of 3 s and intervals of 7 s, totaling 3 min. The process was conducted at 4 °C, followed by centrifugation at 12,000× *g* for 10 min. Subsequently, 0.8 mL of the supernatant was collected, and 0.15 mL of extraction solution II was slowly added with careful blowing and mixing until no gas bubbles were observed. The mixture underwent a 4 °C, 12,000× *g* centrifugation for an additional 10 min, after which the supernatant was collected. The lactic acid content was then determined using the lactic acid content kit by measuring absorbance at 570 wavelengths, and the data were recorded. The lactic acid content was calculated using a standard curve.

### 2.6. Somatic Cell Nuclear Transfer

The CPI-treated PFFs were subjected to digestion and resuspension with DPBS-PVA. In vitro matured oocytes were enucleated through blind aspiration. Blind aspiration is performed with a microfine glass tube under the first polar body to aspirate the first polar body and chromosomes in mid-division and some of the surrounding cytoplasm to ensure removal of the genetic material. Single donor cells were aspirated into the microinjection needle and injected into the zona pellucida of the nucleated oocytes. The reconstructed embryos were then equilibrated in PZM-3 for 1 h. Subsequently, the embryos underwent transfer into an electrofusion solution, and fusion was induced by two direct current (DC) pulses at 100 v/mm for 100 μs. After fusion, they were transferred back into PZM-3 and placed in the incubator. Cleavage rates were observed on the second day of incubation, while blastocysts were collected on the seventh day to record embryonic development efficiency and the number of blastocyst cells.

### 2.7. Transcriptome Sequencing Analysis

Standard transcriptome sequencing was conducted on PFFs subjected to a 50 μM CPI treatment and untreated NC, with four biological replicates for each condition. The treated cells were denoted as CPI 1–4, while the untreated ones were labeled as NC1–4. Total cellular RNA was extracted by the Trizol method as the starting RNA for library construction, and cDNA was synthesized for PCR amplification to construct the library. After checking the quality of the library, Illumina sequencing was performed to obtain clean reads, and HISAT2 (v2.0.5) was used to compare the clean reads with the positional information on the reference genome, count the number of reads for each gene, and calculate the FPKM value. DESeq2 software (1.20.0) was used to perform the statistical analysis to select differentially expressed genes. log2(Foldchange)| > 1 & *p* value < 0.05. The cluster Profiler software (3.8.1) was used to map the DEGs to GO and KEGG databases, and the differential gene sets were analyzed for GO functional annotation and KEGG pathway enrichment. All the above work was conducted by Beijing Novozymes Technology Co., Beijing, China.

### 2.8. Statistical Analysis

The data analysis was conducted using SPSS 20.0, employing *t*-tests for pairwise comparisons between groups, and one-way ANOVA for analyzing differences among two or more groups.

## 3. Results

### 3.1. CPI Treatment Downregulated Oxidative Phosphorylation Gene Expression but Upregulated Glycolytic Enzyme Gene Expression in PFFs

PFFs were treated with different concentrations of CPI (10 μM, 50 μM, and 100 μM) for 48 h. The mRNA expression levels of three oxidative phosphorylation genes (Mitochondrially Encoded Cytochrome C Oxidase I (*COX1*), Mitochondrially Encoded Cytochrome C Oxidase III (*COX3*), Mitochondrially Encoded ATP Synthase Membrane Subunit 6 (*ATP6*)) and three glycolysis genes (Phosphoglycerate Kinase 1 (*PGK1*), Pyruvate Dehydrogenase Kinase 1 (*PDK1*), Lactate Dehydrogenase A (*LDHA*)) in the treated PFFs were measured by qPCR. As depicted in Figure 1, the 10 μM CPI incubation had no significant effect on the expression of all tested genes. The addition of 50 μM CPI to the PFF culture medium significantly decreased *ATP6* and increased *PGK1* mRNA abundance, as compared to the NC group. The supplementation of 100 μM CPI to the culture medium of PFFs significantly downregulated the transcript level of oxidative phosphorylation genes *COX3* and *ATP6* and upregulated the transcript level of glycolytic enzyme gene *PGK1*. These results indicate that treating PFFs with CPI at a concentration of 50 or 100 μM inhibits the expression of oxidative phosphorylation-related genes while promoting the expression of a glycolytic gene, *PGK1*.

### 3.2. CPI Treatment of PFFs Resulted in Downregulation of MMP and Upregulation of Lactate Content

Mitochondria provide energy for the host cells by producing ATPs, predominantly through oxidative phosphorylation. The MMP (Mn) reflects oxidative phosphorylation status of the cells. Therefore, we investigated the effects of CPI treatment on the MMP of PFFs. The results revealed a significant decrease in cellular MMP following the 50 μM and 100 μM CPI treatment, compared to the NC group (Figure 2A,B). To explore the effect of CPI on the PFF glycolytic pathway, we also assessed the lactate content, which is an index of glycolysis, of PFFs treated with 50 μM CPI. The results demonstrated that lactate content in the CPI-treated group was significantly increased, by one-fold, compared to the NC group (Figure 2C). This observation indicates that CPI suppresses the cellular oxidative phosphorylation pathway but promotes the glycolytic pathway, resulting in a drop in cellular MMP and an upregulation of lactate content.

### 3.3. CPI Treatment of Donor Cells PFFs Significantly Enhanced the Developmental Competence of Cloned Embryos

To investigate the influence of CPI treatment of donor cells on subsequent development of porcine cloned embryos, we compared the developmental indexes (cleavage rate, blastocyst rate, and the number of blastocyst cells) of cloned embryos constructed from 50 μM CPI-treated PFFs and NC PFFs. The results revealed that the blastocyst rate (49.33 ± 3.52% vs. 38.00 ± 2.0%, *p* < 0.05) in the treated group was significantly higher than that in the NC group (Table 3). These findings suggest that incubating donor cells with the oxidative phosphorylation inhibitor CPI promotes the subsequent development of cloned pig embryos.

### 3.4. Transcriptomic Analysis of PFFs Treated with CPI

To investigate the impacts of donor cell energy metabolism pathways on the development of porcine cloned embryos, PFFs without any treatment and those treated with 50 μM CPI for 48 h were collected for transcriptome sequencing. Based on the principal component analysis (PCA) results, the NC- and CPI-treated groups exhibited clear grouping, demonstrating significant inter-group differences and intra-group reproducibility (Figure 3A). A total of 10,133 differentially expressed genes (DEGs) were identified between the two groups, comprising 5007 upregulated genes and 5126 downregulated genes in the CPI-treated group (Figure 3B). Gene Ontology (G.O.) and Kyoto Encyclopedia of Genes and Genomes (KEGG) analyses were conducted on the DEGs identified through RNA sequencing. The G.O. analysis revealed enrichment of DEGs in metabolism-related pathways such as the mitochondrial inner membrane protein complex, respirator, inner mitochondrial membrane respiratory chain complex, NADH dehydrogenase activity, oxidoreductase activity and other pathways (Figure 3C). KEGG analysis demonstrated that DEGs between the two cell groups were also enriched in several metabolism-related pathways, including reactive oxygen species, oxidative phosphorylation, thermogenesis, and other pathways. (Figure 3D).

### 3.5. Interaction Analysis of Oxidative Phosphorylation-Related Pathway Networks

The ROS signaling pathway, oxidative phosphorylation pathway, and thermogenic signaling pathway, which are closely related to energy metabolism in the KEGG enrichment analysis, were mapped for network interactions (Figure 4A). The results showed that the key node genes, including cytochrome genes (Mitochondrially Encoded Cytochrome C Oxidase I: *COX1*, Mitochondrially Encoded Cytochrome C Oxidase II: *COX2*, Mitochondrially Encoded Cytochrome C Oxidase III: *COX3*) and ATP synthase genes (Mitochondrially Encoded ATP Synthase Membrane Subunit 6: *ATP6*, Mitochondrially Encoded ATP Synthase Membrane Subunit 8: *ATP8*) had a significantly lower expression in the CPI-treated group than in the NC group (Figure 4B). Cytochrome genes and ATP synthase genes have been shown to inhibit the oxidative phosphorylation pathway [32,33]. The downregulated expression of these two families of genes in the CPI-treated cells indicated that CPI treatment significantly suppressed the oxidative phosphorylation pathways in the cells, further confirming that CPI is an effective oxidative phosphorylation inhibitor. To validate the expression level of five node genes, including *COX1*, *COX2*, *COX3*, *ATP6* and *ATP8*, their mRNA expression levels in 50 μM CPI-treated and NC PPFs were measued by qPCR. As shown in Figure 4C, the expression levels of the tested five genes were lower in the CPI-treated group than in the control group, which was consistent with the downregulation trend in the RNA sequencing results, indicating that the RNA sequencing data were accurate and reliable.

## 4. Discussion

CPI, as an analogue of alpha-lipoic acid, suppresses oxidative phosphorylation by inhibiting the expression of pyruvate dehydrogenase (*PDH*), which is an enzyme complex within the mitochondria pivotal that catalyzes the conversion of pyruvate to acetyl CoA [34,35,36]. The interruption of *PDH* expression by CPI halts the entry of pyruvate into the tricarboxylic acid cycle, disrupting oxidative phosphorylation and diminishing ATP production [37]. Furthermore, the CPI-mediated PDH blockade triggers the accumulation of acetyl-CoA and promotes the glycolytic pathway to meet metabolic requirements [38]. Our qPCR results revealed that the 10 μM CPI treatment exerted no significant effect on the gene expression of energy metabolism pathways in donor cells, whereas the 50 μM and 100 μM CPI supplementation significantly inhibited the expression of oxidative phosphorylation-related genes. Similar results were found in previous research, which demonstrated that a 10 μM CPI treatment failed to stimulate the PFF oxidative phosphorylation pathway, while a higher concentration of 100 μM CPI inhibited oxidative phosphorylation [39].

Inhibition of the oxidative phosphorylation pathway disrupts the electron transport chain’s function in the mitochondria, causing a reduction in the potential gradient between the mitochondrial membranes [40]. In this study, MMP downregulation was observed in 50 μM and 100 μM CPI-treated PFFs. Previous research has shown that 50 μM and 100 μM CPI can decrease the MMP of PFFs [41]. The reprogramming of mouse ear fibroblasts (MEFs) to induce pluripotent stem cells (iPSCs) also involves MMP reduction [42].

During glycolysis, glucose is decomposed into pyruvate, which is then metabolized into lactate in the cytoplasm; therefore, the level of lactate is positively correlated with the intensity of intracellular glycolysis [43]. It has been shown that upregulation of the glycolytic pathway is accompanied by an increase in lactate content [44,45,46]. The significant elevation of lactate levels in the CPI-treated PFFs strongly suggests that the glycolysis process is enhanced in PFFs following CPI treatment.

Among the pathways analyzed by KEGG enrichment in this study, the Wnt, AMPK, and PI3K/AKT signaling pathways are associated with energy metabolism, in addition to the ROS signaling pathway [47], oxidative phosphorylation pathway [45], and thermogenic signaling pathway [48]. It has been shown that activation of the Wnt signaling pathway promotes upregulation of ATP synthase, leading to increased intra-cellular lactate secretion and upregulation of the glycolytic pathway [49]. AMPK enhances the glycolytic pathway by upregulating the expression of glycolytic enzymes [50]. The upregulation of the PI3K/AKT pathway promotes the activities of glycolytic pyruvate kinase and hexokinase, which increase ATP production and promote glycolysis and lactate production in breast cancer cells [51].

In the present study, the blastocyst rate of SCNT embryos cloned from CPI-treated PFFs was significantly higher compared to the NC group treatment of PFFs with 100 µM CoCl_2_ (a glycolysis inducer) and also resulted in a significantly higher blastocyst rate and blastocyst cell number in subsequent SCNT embryos, compared to NC [31]. Similarly, treatment of PFFs with 10 µM PS-48, another glycolysis inducer, led to significantly higher cleavage and blastocyst rates in subsequent cloned embryos compared to NC [30]. These drug treatments all enhance the developmental efficiency of SCNT embryos by altering energy metabolic pathways in donor cells. However, it is important to note that the blastocyst rate and blastocyst cell number may not fully represent the uterine survival rate of cloned embryos [52]. Whether the CPI treatment of donor cells can improve the full-term developmental competence of subsequent cloned embryos remains unknown and requires further investigation.

Our results showed that CPI treatment increased the lactic acid level in donor cells. Lactate has been shown to be essential for the early development of mammalian embryos, and deprivation of lactate leads to significant deletion of H3K18lac and to the failure of major ZAG activation in mouse 2-cell embryos [53]. It has also been shown that lactate promotes epigenetic reprogramming through histone H3K27 acetylation [54]. Increased levels of histone acetylation promote transcriptional activation and increased expression levels of developmentally relevant genes in early SCNT embryos, ultimately enhancing the developmental potential of SCNT embryos [55]. Therefore, CPI treatment improves the developmental efficiency of subsequent cloned embryos.

## 5. Conclusions

Treatment of PFFs with the oxidative phosphorylation inhibitor CPI effectively shifts the cellular energy metabolic pathways from oxidative phosphorylation to glycolysis and enhances the developmental potential of subsequent SCNT embryos. This study provides a simple new method to improve the efficiency of pig cloning, which will be beneficial for promoting the development and application of pig SCNT technology.

## Figures and Tables

**Figure 1 animals-14-01362-f001:**
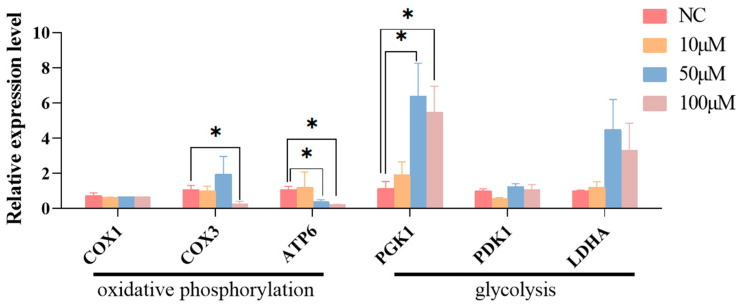
Impacts of CPI treatment on oxidative phosphorylation and glycolysis-related gene expression in PFFs. Data are presented as “Means ± S.E.”. Three biological replicates in each group. “*” indicates a significant difference between two groups in the same column (*p* < 0.05).

**Figure 2 animals-14-01362-f002:**
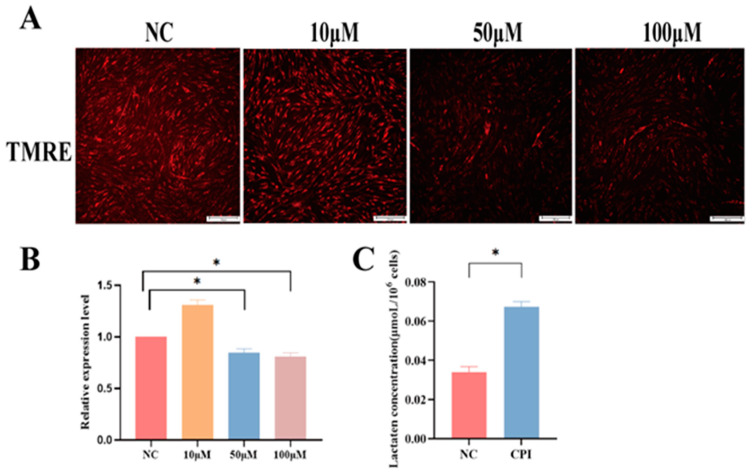
Impacts of CPI treatment on MMP and lactate content in PFFs: (**A**) effects of CPI treatment on MMP of PFFs, PFFs (red) were stained by TMRE. NC represents the negative NC group, PFFs were treated with 10 μM, 50 μM, 100 μM CPI; Original magnification: ×100, bar: 100 μm; (**B**) the relative fluorescence intensity of TMRE probe; and (**C**) effects of 50 μM CPI treatment on lactic acid content in PFFs. Data are presented as “Means ± S.E.”. Three biological replicates in each group. “*” indicates a significant difference between two groups in the same column (*p* < 0.05).

**Figure 3 animals-14-01362-f003:**
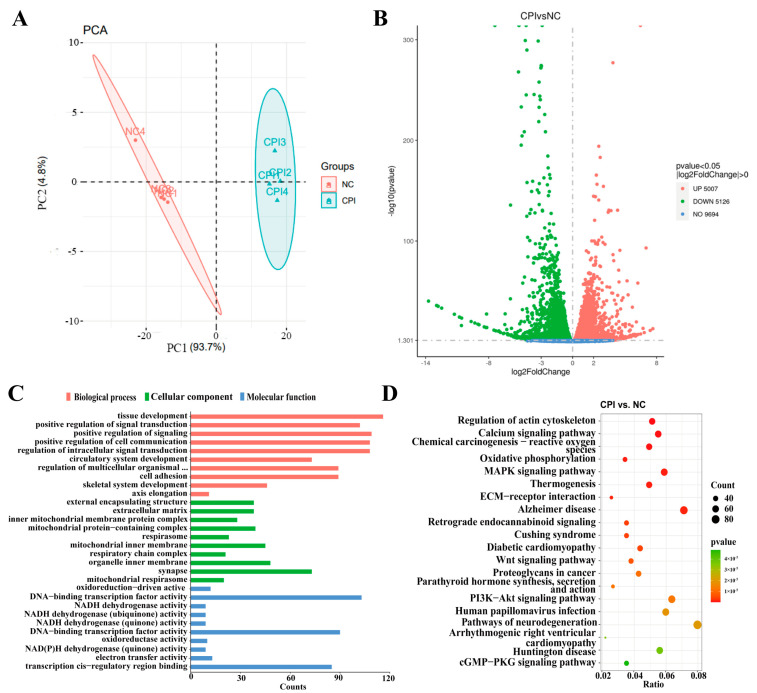
Transcriptome sequencing analysis of CPI-Treated Group and NC Group: (**A**) principal component analysis (PCA) of both groups of cells; (**B**) volcano plot of differentially expressed genes (DEGs) between two groups of cells; (**C**) functional enrichment analysis of DEGs based on the Gene Ontology (G.O.) database, showcasing the top 10 pathways in each subcategory; and (**D**) functional enrichment analysis of DEGs based on the Kyoto Encyclopedia of Genes and Genomes (KEGG), presenting the top 20 KEGG pathways.

**Figure 4 animals-14-01362-f004:**
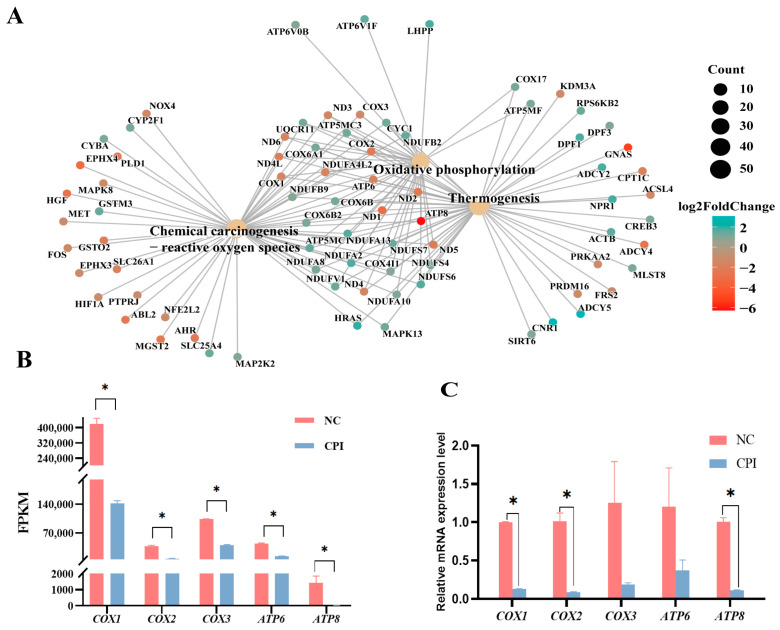
Interaction analysis of oxidative phosphorylation-related pathway networks: (**A**) interaction map of oxidative phosphorylation, ROS, and thermogenic signaling pathway networks; (**B**) comparison of FPKM values of node genes between CPI-Treated Group and NC Group; and (**C**) qPCR validation of RNA sequencing-identified DEGs. Data are presented as “Means ± S.E.”. Three biological replicates in each group. “*” indicates a significant difference between two groups in the same column (*p* < 0.05).

**Table 1 animals-14-01362-t001:** Primer sequence information.

Gene	Primer Sequences (5′–3′)	GenBank ID
β-actin (reference gene)	F: CCTTGGATCTTGGCGGTTCT R: CACTGCCATGCATGATGCTC	NM_001206359
*PGK1*	F: CCTTGGATCTTGGCGGTTCT R: CACTGCCATGCATGATGCTC	NM_001099932
*PDK1*	F: CGTGCTGGGAATCAGCAAAC R: GCTCGAAGTCCGTCTCCTTC	NM_001159608
*LDHA*	F: ATCCTGTGGACGGAAGCATT R: AGGTGATAACAGTGGGTGCG	NM_001172363
*COX1*	F: GGAGGTCTAACGGGCATTGT R: ACCCGGAGAATAGGGGGAAT	NP_008636
*COX2*	F: CCAAGACGCCACTTCACCCATC	NP_008637.1
	R: TGGGCATCCATTGTGCTAGTGTG	
*COX3*	F: AAGACGCCACTTCACCCATC R: TCTTGGGCATCCATTGTGCT	NP_008640
*ATP6*	F: CCGCACAATCTCGATCCAAC R: AGTTGTGTGGTGGGTGTGAA	NP_008639
*ATP8*	F: GCCACAACTAGATCATCCACATG	NP_008638.1
	R: GATTCTGGGCTTGCTGGGTATG	

**Table 2 animals-14-01362-t002:** qPCR reaction system.

Cycle Step	Repetition Number	Temperature	Times
premutability	1	95 °C	30 s
cyclic response	40	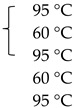	10 s
			30 s
Dissolution curves	1		15 s
			60 s
			15 s

**Table 3 animals-14-01362-t003:** Effects of CPI treatment of donor cells on subsequent development of porcine cloned embryos.

Groups	Cleavage Rate (*n*)	Four-Cell Stage Rate (*n*)	Blastocyst Rate (*n*)	Number of Blastocyst Cells Mean
NC	67.00 ± 1.91% (67/100)	37.00 ± 1.91% (37/100)	38.00 ± 2.0% (38/100)	34.13 ± 13.38
CPI	69.33 ± 1.15% (52/75)	38.67 ± 1.15% (40/75)	49.33 ± 3.52% (43/75) *	43.19 ± 18.52

Data is presented as “Means ± S.E.”. “*” indicates a significant difference between two groups in the same column (*p* < 0.05).

## Data Availability

The data that support the findings of this study are available on request from the corresponding author, upon reasonable request.

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
