# Peer review of "Treatment of Donor Cells with Oxidative Phosphorylation Inhibitor CPI Enhances Porcine Cloned Embryo Development"

_animals, 2024, doi:10.3390/ani14091362_

Round 1

Reviewer 1 Report

Comments and Suggestions for Authors

This study aims to improve the efficiency of somatic cell nuclear transfer by treating donor cells with CPI. The authors sought to elucidate the mechanism of action of CPI through examination of the development rate of cloned embryos, expression of energy metabolism-related genes, transcriptome sequencing analysis, and gene network analysis. The results of the study are promising and could be valuable if applied in the fields of animal reproduction/veterinary medicine. However, some revisions are needed.

1. While gene expression related to energy metabolism was investigated, it's essential to confirm whether glycolysis actually increased in cloned embryos.

2. It would be beneficial to observe embryonic changes such as a reduction in ROS due to decreased oxidative phosphorylation.

Minor:

What was the reason for using blind aspiration in the somatic cell nuclear transfer method? Was it ensured that the nucleus was completely removed after blind aspiration? If such a process exists, please provide additional details.

The significance markings in Figure 1 are not clear. It would be better to extend the lines all the way down or indicate asterisks above the respective columns.

Regarding COX3, is it correct that there was an increase in the 50M treatment group? I suggest verifying experimental procedures for errors or increasing the number of repetitions to reduce errors.

Figure 2A is too dark and blurry to be visible. Please consider enlarging or replacing it with a clearer image.

Author Response

Dear Reviewers and editors,

Thank you very much for reviewing our manuscript and giving constructive and helpful comments. We have revised and improved our manuscript following your comments. The revision was marked in the revised manuscript. In the following please find our point by point response to your comments.

Review comments

Reviewer 1

  1. While gene expression related to energy metabolism was investigated, it's essential to confirm whether glycolysis actually increased in cloned embryos.

Response: Thank you very much for your comments and suggestions. The main purpose of this study was to enhance cloned embryo development by changing the energy metabolism status of donor cells (not cloned embryos) via treating donor cells with CPI, an inhibitor of oxidative phosphorylation. Our results showed that the expression level of genes related to glycolysis and the lactic acid content, which is a product of glycolysis, were increased in CPI-incubated donor cells. These results strongly suggest that glycolysis was enhanced in donor cells treated with CPI. However, treatment of donor cells with CPI does not necessarily increase glycolysis in cloned embryos, because cloned embryos were not treated with CPI in this study. CPI treatment changed the energy metabolism status of donor cells, which then probably altered the epigenetic modification of donor cells and thereby improving the development of subsequent cloned embryos. Therefore, the energy metabolism may not affected by CPI treatment of donor cells. This was the reason why we did not measure glycolysis in cloned embryos.

  1. It would be beneficial to observe embryonic changes such as a reduction in ROS due to decreased oxidative phosphorylation.

Response: The main purpose of this study was to decrease oxidative phosphorylation in donor cells but not cloned embryos. The CPI drug was used to treat donor cells but not cloned embryos. Our results showed that the expression level of genes related to oxidative phosphorylation and the mitochondrial membrane potential (MMP), which is an indicator of oxidative phosphorylation, were decreased in CPI-incubated donor cells. These results strongly suggest that oxidative phosphorylation was inhibited in donor cells treated with CPI. However, treatment of donor cells with CPI does not necessarily reduce oxidative phosphorylation in cloned embryos. Therefore, we did not evaluate oxidative phosphorylation in cloned embryos.

Minor:

1.What was the reason for using blind aspiration in the somatic cell nuclear transfer method? Was it ensured that the nucleus was completely removed after blind aspiration? If such a process exists, please provide additional details.

Response: The SCNT operators in our lab have over 13 years experience in constructing cloned embryos. Our previous data showed that blind aspiration conducted in our lab results in an oocyte enucleation rate at about 93% (evaluated by staining enucleated oocytes with DNA dye). Blind aspiration-based oocyte enucleation method results in a high enucleation rate, but with a lower damaging effect than other methods such as DNA dye staining-assisted or spindle imaging system-assisted enucleation. Therefore, our lab chose to use blind aspiration method for oocyte enucleation. We already provided additional details about this method in the Methods and Materials section of our revised manuscript.

2.The significance markings in Figure 1 are not clear. It would be better to extend the lines all the way down or indicate asterisks above the respective columns.

Response: Thank you very much for your suggestions. We have changed Figure 1 following your suggestion.

3.Regarding COX3, is it correct that there was an increase in the 50M treatment group? I suggest verifying experimental procedures for errors or increasing the number of repetitions to reduce errors.

Response:  The Figure 1 in the first version of our manuscript did not show that COX3 expression was significantly increased in the 50 μM CPI treatment group. To reduce errors, the data of each group was measured from three independent samples and each sample was measured by three technical replicates.

4.Figure 2A is too dark and blurry to be visible. Please consider enlarging or replacing it with a clearer image.

Response:  Thank you very much for your suggestions. The Figure 2A in the manuscript have been modified following your suggestion

Reviewer 2 Report

Comments and Suggestions for Authors

This mauscript covers the effect of CPI, an oxidative phosphorylation inhibitor, on energy metabolism pathways of porcine fibroblasts and development of subsequent embryos obtained with somatic cell nuclear transfer (SCNT) technology. Authors investigated the influence of CPI on porcine fetal fibroblasts (PFFs), donor cells for cloned embryos. The energy metabolic status, global gene expression patterns and the developmental competence of SCNT embryos were compared between CPI-treated PFFs and PFFs non-treated with CPI. Results preseted in this study revealed that in CPI-treated PFFs the expression of oxidative phosphorylation-related genes was inhibited, while expression of glycolitic gene was enhanced. Authors  observed that CPI treatment of PFFs resulted in down-regulation of MMP and up-regulation of lactate content, which means that CPI suppresses the cellular oxidative phosphorylation pathway and promotes the glycolytic pathway. It was also demonstrated that CPI treatment of donor cells PFFs increased the blastocyst rate, which suggests that this molecule promotes following development of cloned embryos. Transcriptomic analysis of CPI-treated PFFs showed 10,133 differentially expressed genes including genes related to metabolism. Taken together, CPI treatment of PFFs changes the cellular energy metabolic pathway from oxidative phosphorylation to glycolysis and increases the developmental potential of subsequent SCNT embryos.

The manuscript was quite interesting to read, however I would like to make  few comments, which are listed below. I would appreciate if authors could respond to them.

Line 92: The abbreviation NC is not explained. I understand this is a control, but the abbreviation should be clearly explained when first used in the text.

Line 109: On what basis the CPI doses used in the experiment  were selected?

Lines 110- 118: Why did you use only one reference gene? Did you consider using geometric means of two references genes? What were the reaction conditions for each examined gene? Did you prepare samples in duplicates? What type of negative control was used? How did you confirm the specificity of amplication and what method was used to calculate relative expression of tested genes? I think that adding information about the conditions of the qPCR reaction to the methodology will enrich the manuscript and would be helpful in reproducing the experiment by other researchers. 

Line 120: On what basis was the incubation time with CPI selected? 

Line 128: Please provide the exact name of the software.

Line 150: What does D.C. mean? Please explain this abbreviation.

Line 176: It is not clear to me how the results are presented. Are they presented as means  ± standard error? How many biological replicates were conducted? It should be point out in at least in Figure caption. In my opnion Figures should be bigger, so it would be easier to read them. 

Lines 189-191: "(...) These results indicate that treating PFFs with CPI at a concentration of 50 or 100 µM inhibits the expression of oxidative phosphorylation-related genes while promoting the expression of glycolytic genes"

As far as I can see in the Figure 1, CPI treatment affected only one glycolysis gene, which is PGK1.

Line 196: The same comment as in the line 176.

Lines 229 and 253: Figures in that sections are too small.

Line 229: Have at least few selected DEGs been validated using real-time PCR?

Line 271: Discussion section is described quite briefly. The manuscript would benefit if it were expanded of more information.

Line 308- 309: the sentence is grammatically incorrect.

Author Response

Reviewer 2

The manuscript was quite interesting to read, however I would like to make few comments, which are listed below. I would appreciate if authors could respond to them.

Response: Thank you very much for your comments.

1.Line 92: The abbreviation NC is not explained. I understand this is a control, but the abbreviation should be clearly explained when first used in the text.

Response: Thank you very much for your suggestions. Changes have been made in the text following your suggestion.

2.Line 109: On what basis the CPI doses used in the experiment were selected?

Response: The concentration of CPI was chosen according to the following literature: Mordhorst BR, Murphy SL, Ross RM, Benne JA, Samuel MS, Cecil RF, Redel BK, Spate LD, Murphy CN, Wells KD, Green JA, Prather RS. Pharmacologic treatment of donor cells induced to have a Warburg effect-like metabolism does not alter embryonic development in vitro or survival during early gestation when used in somatic cell nuclear transfer in pigs. Mol Reprod Dev. 2018 Apr;85(4):290-302. doi: 10.1002/mrd.22964. Epub 2018 Mar 5. PMID: 29392839; PMCID: PMC5903921.

3.Lines 110- 118: Why did you use only one reference gene? Did you consider using geometric means of two references genes? What were the reaction conditions for each examined gene? Did you prepare samples in duplicates? What type of negative control was used? How did you confirm the specificity of amplication and what method was used to calculate relative expression of tested genes? I think that adding information about the conditions of the qPCR reaction to the methodology will enrich the manuscript and would be helpful in reproducing the experiment by other researchers.

Response: Many studies have reported that only one reference gene was used in qPCR experiment. In this study, qPCR was only performed on one type of sample, pig fibroblasts. We believe that use of one reference gene can obtain reliable data. We have added more details about the qPCR reaction conditions for each examined gene to the revised manuscript.  The data of each group was measured from three independent samples and each sample was measured by three technical replicates. We did not used negative control in this study. A melting curve analysis was added at the end of the PCR amplification procedure to confirm the specificity of amplication. We used the 2-∆∆Ct method to calculate relative expression of tested genes. Detailed information about the conditions of the qPCR reaction has been added to the methodology of our revised manuscript.

4.Line 120: On what basis was the incubation time with CPI selected?

Response:  We chose to use 48 hours as CPI incubation time because the CPI-treated pig fibroblasts usually reach a full confluence at about 48 hours after CPI treatment.

5.Line 128: Please provide the exact name of the software.

Response: The ImageJ quantized images software was used. We have added this information to our revised manuscript.

6.Line 150: What does D.C. mean? Please explain this abbreviation.

Response: D.C. should be direct current (DC) pulses. We already changed it into direct current (DC) pulses in the manuscript.

7.Line 176: It is not clear to me how the results are presented. Are they presented as means ± standard error? How many biological replicates were conducted? It should be point out in at least in Figure caption. In my opnion Figures should be bigger, so it would be easier to read them.

Response:  Data is presented as "Means ± S.E." Each group has three biological replicates." This information has been added to the Figure caption, and the Figures were modified to make them easier to read.

8.Lines 189-191: "(...) These results indicate that treating PFFs with CPI at a concentration of 50 or 100 µM inhibits the expression of oxidative phosphorylation-related genes while promoting the expression of glycolytic genes. As far as I can see in the Figure 1, CPI treatment affected only one glycolysis gene, which is PGK1.

Response: Thank you for pointing out this mistake for us. The sentences have been corrected in revised manuscript.

9.Line 196: The same comment as in the line 176.

Response:  Data is presented as "Means ± S.E." Each group has three biological replicates." This information has been added to the Figure caption, and the Figures were modified to make them easier to read.

10.Lines 229 and 253: Figures in that sections are too small.

Response: The figures and the font size have been adjusted in the manuscript to make them easied to read.

11.Line 229: Have at least few selected DEGs been validated using real-time PCR?

Response:  Thank you for your suggestion. Five genes were selected for validation by qPCR. The qPCR results match with those measured by RNA sequencing. The qPCR Validation result has been added to the revised manuscript (see Figure 4c).

12.Line 271: Discussion section is described quite briefly. The manuscript would benefit if it were expanded of more information.

Response: Following your suggestion, we have added one paragraph to the Discussion section in revised manuscript.

13.Line 308- 309: the sentence is grammatically incorrect.

Response: Thank you for pointing out this mistake for us. The sentence has been corrected in revised manuscript.

Round 2

Reviewer 2 Report

Comments and Suggestions for Authors

I would like to thank the authors for preparing a response to my comments. I have a few suggestions which I listed below.

You stated that CPI doses were chosen based on this article: Mordhorst BR, Murphy SL, Ross RM, Benne JA, Samuel MS, Cecil RF, Redel BK, Spate LD, Murphy CN, Wells KD, Green JA, Prather RS. Pharmacologic treatment of donor cells induced to have a Warburg effect-like metabolism does not alter embryonic development in vitro or survival during early gestation when used in somatic cell nuclear transfer in pigs. Mol Reprod Dev. 2018 Apr;85(4):290-302. doi: 10.1002/mrd.22964. I suggest to add this reference to the manuscript in the Methodology section.

Line 118: I suggest to add this reference to the manuscript: Livak KJ, Schmittgen TD. Analysis of relative gene expression data using real-time quantitative PCR and the 2(-Delta Delta C(T)) Method. Methods. 2001 Dec;25(4):402-8. doi: 10.1006/meth.2001.1262. PMID: 11846609.  I also need to emphasize that negative controls (using RNase-free water instead of cDNA) should be performed during each experiment, to make sure that reagents were not contaminated before starting the experiment.

Line  161: As far as I can see D.C. abbreviation is still not explained in the text.

Line 300: You should point out in the Figure 4C significant differences between groups and include in the Figure caption how the data was presented.

Lines 340-341: the following sentence is still incorrect. In my opinion it should be as follows: In the present study, the blastocyst rate of SCNT embryos cloned from CPI -treated PFFs was significantly higher compared to the NC group.

Line 359: I think the correct version of this sentence is as follows: “Therefore, CPI treatment improves the developmental efficiency of subsequent cloned embryos”.

My general suggestion is to have your manuscript proofread by a native English speaker.

Author Response

Dear Reviewers and editors,

Thank you very much for reviewing our manuscript and giving constructive and helpful comments. We have revised and improved our manuscript following your comments. The revision was marked in the revised manuscript. In the following please find our point by point response to your comments.

Reviewer comments:

1.You stated that CPI doses were chosen based on this article: Mordhorst BR, Murphy SL, Ross RM, Benne JA, Samuel MS, Cecil RF, Redel BK, Spate LD, Murphy CN, Wells KD, Green JA, Prather RS. Pharmacologic treatment of donor cells induced to have a Warburg effect-like metabolism does not alter embryonic development in vitro or survival during early gestation when used in somatic cell nuclear transfer in pigs. Mol Reprod Dev. 2018 Apr;85(4):290-302. doi: 10.1002/mrd.22964. I suggest to add this reference to the manuscript in the Methodology section.

Response: Thank you very much for your suggestions. This reference has been cited in the Methods and Materials section of our revised manuscript.

2.Line 118: I suggest to add this reference to the manuscript: Livak KJ, Schmittgen TD. Analysis of relative gene expression data using real-time quantitative PCR and the 2(-Delta Delta C(T)) Method. Methods. 2001 Dec;25(4):402-8. doi: 10.1006/meth.2001.1262. PMID: 11846609.  I also need to emphasize that negative controls (using RNase-free water instead of cDNA) should be performed during each experiment, to make sure that reagents were not contaminated before starting the experiment.

Response: Thank you very much for your suggestions. Thereference that you recommended has been added to the manuscript.

3.Line 161: As far as I can see D.C. abbreviation is still not explained in the text.

Response: Thank you very much for your comments. This sentence has been revised in the manuscript.

4.Line 300: You should point out in the Figure 4C significant differences between groups and include in the Figure caption how the data was presented.

Response: Thank you very much for your comments and suggestions. Significance between the two groups was added to Figure 4C and the figure caption has been revised to describe how the data was presented.

5.Lines 340-341: the following sentence is still incorrect. In my opinion it should be as follows: In the present study, the blastocyst rate of SCNT embryos cloned from CPI -treated PFFs was significantly higher compared to the NC group.

Response: Thank you very much for your comments and suggestions.The sentence has been revised following your suggestion.

6.Line 359: I think the correct version of this sentence is as follows: “Therefore, CPI treatment improves the developmental efficiency of subsequent cloned embryos”.

Response: Thank you very much for your comments and suggestions. The sentence has been revised following your suggestion.